# The Roles of Obesity and ASB4 in Preeclampsia Pathogenesis

**DOI:** 10.3390/ijms25169017

**Published:** 2024-08-20

**Authors:** Yuye Wang, Rebecca Ssengonzi, W. H. Davin Townley-Tilson, Yukako Kayashima, Nobuyo Maeda-Smithies, Feng Li

**Affiliations:** Department of Pathology and Laboratory Medicine, The University of North Carolina, Chapel Hill, NC 27599, USA; yuye@email.unc.edu (Y.W.);

**Keywords:** preeclampsia, ASB4, obesity, insulin, leptin, implantation, endometrium, trophoblast

## Abstract

Preeclampsia is a complex pregnancy-related hypertensive disorder which poses significant risks for both maternal and fetal health. Preeclampsia affects 5–8% of pregnancies in the United States, causing a significant public health and economic burden. Despite extensive research, the etiology and pathogenesis of preeclampsia remain elusive, but have been correlated with maternal conditions such as obesity. In recent decades, the incidence of preeclampsia increased along with the prevalence of obesity among women of reproductive age. Maternal obesity has been shown to negatively affect pregnancy in almost all aspects. However, the precise mechanisms by which obesity influences preeclampsia are unclear. Ankyrin repeat and SOCS Box Containing protein 4 (ASB4) is an E3 ubiquitin ligase that can promote the degradation of a wide range of target proteins. ASB4-null mice display a full spectrum of preeclampsia-like phenotypes during pregnancy including hypertension, proteinuria, and decreased litter size. Furthermore, maternal obesity induced by a high-fat diet aggravates preeclampsia-like phenotypes in pregnant mice lacking ASB4. Variants in the ASB4 gene have been associated with obesity in humans, and a functional connection between the ASB4 gene and obesity has been established in mice. This review discusses the connections between preeclampsia, obesity, and ASB4.

## 1. Introduction

Preeclampsia is a hypertensive disorder unique to pregnancy that occurs in approximately 5–8% of U.S. pregnancies [1]. In the United States, incidences of preeclampsia have dramatically increased in the past three decades [2,3]. Preeclampsia usually occurs after 20 weeks of gestation and is characterized as new onset hypertension, proteinuria, and end-organ damage. While the pathogenesis of preeclampsia is not fully clear, it is believed to originate in the placenta. The alterations in trophoblast invasion result in insufficient placental perfusion and the subsequent release of pro-inflammatory and anti-angiogenic factors into the maternal circulation, which damages maternal endothelial function [4]. Maternal obesity is a major risk factor for preeclampsia, and although the relationship between obesity and preeclampsia has been established in human studies and experimental animal studies, the precise mechanism by which maternal obesity influences preeclampsia is unknown. While obesity-related inflammation, oxidative stress, insulin resistance, and underlying hypertension have been suggested, at what stage(s) and how these adverse effects of maternal obesity influence the development of preeclampsia is incompletely understood [5]. Ankyrin-repeat-and-SOCS-box-containing-protein 4 (ASB4) is a component of the E3 ubiquitin ligase, and ASB4 is critical during early vascular development and proper placentation [6,7]. ASB4-null female mice develop a full spectrum of phenotypes of preeclampsia during late-stage pregnancy including hypertension, proteinuria, and decreased litter size [7,8]. High-fat diet (HFD)-induced obesity aggravates preeclampsia-like phenotypes in ASB4-null female mice [9]. In this review, we discuss the potential roles of altered metabolic factors on preeclampsia pathogenesis in obese mice lacking ASB4.

## 2. Preeclampsia

There are two major classifications of preeclampsia: early-onset preeclampsia (EOP), which develops before 34 weeks of pregnancy, and late-onset preeclampsia (LOP), which develops on or after 34 weeks of pregnancy. About 20% of preeclampsia cases are EOP, which is most commonly associated with severe clinical symptoms, while patients with LOP typically have milder symptoms [10]. Preeclampsia is a leading cause of maternal morbidity and mortality; women who develop preeclampsia are at an increased risk of pulmonary edema [11], coagulation defects, hepatic and/or renal failure, seizures, cerebral hemorrhage, blindness [12,13,14], and even death [15]. Preeclampsia is also a significant contributor to neonatal morbidity and mortality. Fetal growth restriction—likely due to chronic placental hypoperfusion—is a common complication of preeclampsia [16]. Fetal growth restriction makes women with preeclampsia 3–4 times more likely to deliver small-for-gestational-age babies compared to women with normal pregnancy [17]. Placental abruption, the premature separation of the placenta with a disruption of blood flow to the fetus, is an uncommon but dangerous complication that occurs in up to 3% of pregnancies complicated by severe preeclampsia [18]. Preeclampsia not only affects women and their babies during pregnancy but also in the years following pregnancy. Preeclamptic mothers and their offspring have an increased risk of developing cardiovascular and cognitive problems, among others, in later life [19,20,21].

At present, there are few measures to predict or prevent the development of preeclampsia. The only definitive treatment for preeclampsia is the delivery of the placenta and infant as physicians did a century ago [22]. During pregnancy, treatments for preeclampsia include the monitoring of maternal and fetal health, as well as administering non-teratogenic anti-hypertensive drugs such as methyldopa to lower blood pressure and an anti-seizure drug, magnesium sulfate, to lower heart rate and prevent eclampsia, which is a rare, but life-threatening condition that is characterized by the onset of seizures or coma [23,24]. Treatment other than delivery is an unmet need for women with preeclampsia.

While the etiology and pathogenesis of preeclampsia remain unclear, defective placentation resulting from impaired trophoblast invasion is thought to be at the center of preeclampsia pathogenesis (Figure 1) [25]. Impaired trophoblast invasion during early pregnancy leads to the insufficient remodeling of maternal spiral arteries which subsequently decreases blood flow to the placenta [26]. The ischemic placenta releases anti-angiogenic and inflammatory factors such as sFlt-1 (soluble fms-like tyrosine kinase 1, a soluble form of vascular endothelial growth factor receptor 1) into the maternal circulation, causing maternal endothelial dysfunction and clinical manifestations of preeclampsia: hypertension, proteinuria, and end-organ damage [27]. Pre-pregnancy health conditions including obesity, chronic hypertension, diabetes mellitus, renal disease, and autoimmune disorders increase the risk of preeclampsia [28]. These pre-existing maternal conditions can impact the severity of preeclampsia by negatively affecting placentation and the maternal endothelium.

## 3. Obesity and Preeclampsia

Obesity is defined as excessive fat accumulation that presents a risk to health [29]. The incidence of obesity has continued to increase among women of reproductive age, causing approximately 50% of women worldwide to enter pregnancy overweight or obese [30,31]. Obesity is a major risk factor for preeclampsia [5]. Over the past three decades, the increase in preeclampsia reflected the increased prevalence of obesity in the United States [32,33], as the rate of preeclampsia increased with increasing body mass index (BMI) [34]. Even in non-obese women, the occurrence of preeclampsia is positively correlated with BMI [35]. Obesity increases the risk of all types of preeclampsia, namely, severe and mild preeclampsia, as well as EOP and LOP [5]. Obesity is also associated with maternal hyperinsulinemia, increased inflammation and oxidative stress, and altered adipokines and pro-/anti-angiogenic factors [36,37,38,39], which also contribute to preeclampsia pathogenesis (Figure 2). In the following sections, we will discuss the role of insulin and leptin in preeclampsia as these factors are altered in the obese mouse model of preeclampsia—i.e., feeding ASB4-null mice with a high-fat diet (HFD), as described in the last section of this review.

### 3.1. Insulin

#### 3.1.1. Insulin and Implantation

Insulin is a polypeptide hormone secreted by the pancreas that has a role in glucose, lipid, and protein metabolism [40]. In the uterus, insulin signaling is essential for endometrial epithelial and stromal cell proliferation and differentiation before implantation [41]. Implantation is a delicate interaction between the embryo and endometrium, and it is crucial for the establishment of a successful pregnancy. Altered endometrial receptivity and/or trophoblast cell biology lead to impaired implantation and insufficient placentation [42]. Insufficient placentation subsequently causes the release of anti-angiogenic and pro-inflammatory factors into the maternal circulation which contribute to preeclampsia development, especially in EOP [43].

#### 3.1.2. Excess Insulin in Maternal Endometrial Receptivity

Hyperinsulinemia is an established characteristic of obesity [44,45]. Insulin may impair implantation by interfering with endometrial receptivity: insulin resistance does not affect early embryo development but decreases the implantation rate in the in vitro maturation–in vitro fertilization–embryo transfer cycle of women with polycystic ovarian syndrome (PCOS)—a syndrome associated with insulin resistance [46]. Women with PCOS have endometrial receptivity problems such as impaired decidualization and placentation, as well as an increased risk of preeclampsia [47]. Animal studies demonstrate that insulin signaling is essential for normal endometrial structure and function [41]; however, excess insulin has adverse effects on endometrial receptivity. For example, Li R. et al. reported that the exogenous administration of insulin prior to mating induced hyperinsulinemia in wild type (WT) female mice [48]. The expression of markers of endometrial receptivity (i.e., estrogen receptor, progesterone receptor, and homeobox A10) were altered in these mice, leading to the conclusion that endometrial receptivity in these mice was impaired by maternal hyperinsulinemia [48]. However, the authors did not observe a decreased number of implantation sites as Li M. et al. observed in mice with poor endometrial receptivity caused by haploinsufficiency for adrenomedullin, which has been established as a mouse model that exhibits reduced maternal fertility with endometrial receptivity [49]. Furthermore, Li R. et al. did not identify any obvious morphological changes in the endometrium of pregnant WT mice with an exogenous administration of insulin, examined using light microscopy. Whether there is ultra-structural change is unclear [49]. 

High doses of insulin (5–500 nM) increased prokineticin expression during the decidualization of human endometrial stromal cells, inhibiting the migratory and invasive capacity of trophoblast cells [50]. Supporting the hypothesis that insulin adversely affects endometrial receptivity, our unpublished data showed that Ishikawa cells, an epithelial cell line derived from an endometrial adenocarcinoma, had decreased normal trophoblast spheroid attachment to the cell monolayer after treatment with a high dose of insulin (500 nM). Overall, insulin could cause the dysfunction of endometrial epithelial and stromal cells, leading to impaired endometrial receptivity. 

#### 3.1.3. Excess Insulin in Trophoblast Cells 

Embryo-derived trophoblasts also play an important role in implantation, but it is almost impossible to study trophoblast biology at such an early stage of human pregnancy due to ethical issues. Cell culture models and experimental animal studies provide insights on the effect of excess insulin in trophoblast cells. Vega et al. reported that human primary first trimester trophoblast cells exposed to 1 nM of insulin (the human physiological concentration of insulin is 0.1 nM) in culture had decreased cell survival due to increased DNA damage and apoptosis—as evidence by increased γ-H2AX (a DNA damage marker) and cleaved caspase-3 (a marker of apoptosis) [51]. Furthermore, pre-treatment with metformin (a drug which improves insulin sensitivity by increasing peripheral glucose uptake and utilization) prevented the detrimental effects of insulin [51]. 

In the immortalized human first trimester trophoblast cell line HTR8/SVneo, Silva et al. found that insulin (1–10 nM for 48 h) decreased cell proliferation in a dose- and time-dependent manner, possibly in association with the activation of downstream mediators of insulin including phosphoinositide 3 kinase (PI3K), mammalian target of rapamycin (mTOR), and p38 mitogen-activated protein kinase (p38 MAPK) [52]. Quercetin is a flavonoid occurring in food and is also available as a supplement. Simvastatin is a statin commonly used in clinical practice to lower cholesterol. It was reported that these two chemicals, when treated separately, inhibited insulin’s antiproliferative effect [52]. In contrast, exposing trophoblast cells to insulin (10 nM for 48 h) showed no significant effect on cell viability, apoptosis, or migration capacity [52].

Later, a publication reported that by treating HTR-8/SVneo cells with a higher dose of insulin (1 µM) for 48 h, the proliferation, migration, and invasion of cells were suppressed [53]. Taken together, these in vitro data suggest that higher-than-normal insulin negatively affects trophoblast cells and could lead to impaired implantation. More study is needed to provide strong evidence to support the theory that insulin directly impairs trophoblast cells and subsequent implantation.

#### 3.1.4. Insulin in Maternal Endothelium

Hyperinsulinemia may also aggravate the effects of placenta-derived anti-angiogenic/inflammatory factors on pre-existing endothelial problems. Under normal physiological conditions, insulin maintains the endothelial health by balancing the production of vasodilator [endothelial nitric oxide synthase (eNOS)-derived nitric oxide (NO)] and vasoconstrictor (endothelin-1, ET-1) using two separate pathways. Insulin increases eNOS activity and NO production through the insulin receptor (IR)/insulin receptor substrate 1 (IRS-1)/PI3K/phosphoinositide-dependent kinase-1 (PDK-1)/Ak strain transforming (AKT) pathway [54], while insulin stimulates ET-1 secretion through the IR/growth factor receptor-bound protein 1(GRB-1)/rat sarcoma (RAS)/rapidly accelerated fibrosarcoma (RAF)/MAPK pathway [54]. Under hyperinsulinemia conditions, the PI3K pathway in vascular tissues is seriously impaired, whereas the MAPK pathway remains unaffected, leading to increased ET-1/decreased NO production [55,56]. This imbalance caused by hyperinsulinemia could be exaggerated by detrimental factors derived from placentas in preeclamptic pregnancies. 

### 3.2. Leptin

#### 3.2.1. Leptin and Implantation

Leptin—a 167-amino acid (16 kDa) peptide hormone mainly secreted by adipose tissue—regulates food intake, body mass, and reproductive function [57,58]. While leptin is required before and during implantation, it is not essential for late pregnancy and parturition (giving birth) [59]. Obesity causes hyperleptinemia (elevated plasma leptin levels) in humans and mice [60,61,62], which is associated with preeclampsia [63,64]. Low plasma leptin during the first trimester has been associated with women who suffer spontaneous first trimester loss, implying the direct role of leptin in implantation [65]. Animal studies also provide clear evidence of the indispensable role of leptin during implantation. Female mice lacking leptin are infertile; however, the administration of leptin (5 mg/kg twice daily) only during implantation—i.e., the first 6 days of pregnancy—leads to normal pregnancy and parturition [59]. 

#### 3.2.2. Excess Leptin in Maternal Endometrial Receptivity 

PCOS patients with elevated serum leptin levels had increased very early pregnancy loss and a decreased expression of the γ epithelial Na^+^ channel (γ-ENaC, important in embryo implantation) in the secretory phase of the endometrium [66]. In a cell culture model, treatment with high doses of leptin (50–200 ng/mL) downregulated the expression of γ-ENaC, which was accompanied with decreased trophoblast spheroid attachment to cultured endometrial epithelial cells (Ishikawa cells) [66]. The knockdown of STAT3 (signal transducer and activator of transcription 3), a mediator of leptin action, blocked the effects of leptin [66]. in vitro studies reported that leptin inhibits normal human endometrial stromal cell differentiation, which is critical for proper endometrial receptivity [67]. Taken together, these data suggest that excess leptin/STAT3 signaling may impair both endometrial epithelial and stromal cell function, leading to defective endometrial receptivity. 

#### 3.2.3. Excess Leptin in Trophoblast Cells

Though there are several studies that provide information on the role of leptin in trophoblasts, the direct effects of excess leptin on trophoblasts are not clear [68]. Using the human first trimester placenta-derived extravillous trophoblast cell line (TEV-1) [69], Liu et al. found that 5–500 ng/mL of leptin (non-obese and non-pregnant adult women have serum leptin levels around 8 ng/mL) decreased cell proliferation after 24 h of treatment. The highest dose of leptin treatment promoting trophoblast invasion and migration [70,71], indicating that excess leptin could inhibit trophoblast proliferation but promote differentiation. Two other groups have similarly reported that leptin stimulated the invasion of another trophoblast cell line (HTR8/SVneo) [72,73]. Mechanistically, it was proposed that the crosstalk between the metastasis associated 1 (MTA1)/wingless-related integration site (WNT) and PI3K/AKT pathways possibly increases matrix metallopeptidase 14 (MMP-14) [72]. On the other hand, Wang et al. postulated that the crosstalk between the neurogenic locus notch homolog protein 1 (NOTCH1) and PI3K/AKT pathways activates β-catenin, thereby driving trophoblast invasion [73]. Trophoblast progenitors must sufficiently proliferate before differentiating into distinct trophoblast cell lineages. If leptin disrupts the proliferative process, trophoblasts could differentiate prematurely and negatively affect embryo implantation.

#### 3.2.4. Leptin in Maternal Endothelium

In obese mice induced by HFD (60% calories from fat), Korda et al. found a 2.5-fold increase in circulating leptin concentration [74]. This increase coincides with a reduction in bioavailable NO and elevated concentrations of superoxide and peroxynitrite, both of which are due to eNOS uncoupling. They also found that in endothelial cells (HUVECs), exposure to leptin (0.1 µM for 12 h) increased eNOS expression and NO production, and dramatically increased superoxide and peroxynitrite levels [74]. The authors concluded that hyperleptinemia-induced endothelial dysfunction is mediated by a redox imbalance resulting from eNOS uncoupling. Manuel-Apolinar et al. reported that leptin mediated an increased expression of intercellular adhesion molecules (e.g., ICAM-1) and cyclooxygenase 2 in isolated rat aorta tissue, leading to endothelial dysfunction [75].

Taken together, leptin directly affects endothelial function through multiple molecular pathways, and these altered molecules/pathways could exacerbate the endothelial dysfunction caused by placental soluble factors, worsening the symptoms of preeclampsia.

## 4. ASB4

Ankyrin Repeat and SOCS Box-Containing protein 4 (ASB4) was initially discovered in mice as an imprinted gene that is only maternally expressed [76]. Like most proteins in the ASB4 family, ASB4 acts as an adaptor for the subunits of cullin-based ubiquitin ligases [77]. ASB4 is a substrate-recognition component of the E3 ubiquitin ligase complex [6]. Nine N-terminal ankyrin repeats of ASB4 serve as the substrate-binding site, and the C-terminal suppressor of cytokine signaling (SOCS) box interacts with elongin B/C, rendering the E3 ligase function [77,78,79,80]. Together, the E3 ligase complex mediates the ubiquitination and subsequent proteasomal degradation of the target proteins [78]. 

### 4.1. ASB4 and Obesity

Genome-wide association studies have identified the link between single nucleotide polymorphisms (SNPs) in the ASB4 gene and obesity [81,82]. ASB4’s specific expression in the hypothalamic arcuate nucleus (ARC) regulates energy homeostasis and food intake through insulin and leptin [81,82]. Leptin increases ASB4 expression in the ARC while insulin seems to have no effect [83]. During fasting, ASB4 expression is decreased in the hypothalamus, and it is indirectly downregulated by Agouti-related protein (AgRP), an orexigenic neuropeptide [84,85]. ASB4 is expressed in proopiomelanocortin (POMC) and neuropeptide Y (NPY) neurons in the ARC [81,83]. In mice, POMC-specific ASB4 knockdown resulted in impaired glucose tolerance without obesity, while overexpressing ASB4 in POMC neurons caused an increase in food intake without obesity [84]. ASB4 is also required in calcitonin signaling. Global ASB4 deficiency in mice leads to the decreased expression of calcitonin receptor (CalcR), which inhibits the anorectic effect of calcitonin, and only male mice, not female mice, lacking ASB4 started becoming heavier than WT mice at 5 months of age [84]. 

In human subjects with type 2 diabetes, ASB4 expression is reduced in the hypothalamus, suggesting the potential role of ASB4 in satiety and glucose homeostasis [84]. Mechanistic studies have reported that ASB4 colocalizes with insulin receptor substrate 4 (IRS4) in the hypothalamus [81]. ASB4 increases IRS4 ubiquitination and decreases the phosphorylation of AKT, thereby reducing the downstream signaling of IRS4 and potentially regulating the action of insulin and leptin in the hypothalamus [81]. ASB4 also interacts with G-protein pathway suppressor 1 (GPS1), and this interaction decreases the phosphorylation of IRS1 and inhibits c-Jun NH_2_- terminal kinase (JNK) activity [86]. 

ASB4’s involvement in cancer cell lines might also provide potential links with obesity. In liver cancer, ASB4 expression is upregulated by microRNA-200a (miRNA-200a), and suppressing ASB4 mitigated the invasion and migration ability of a subset of hepatocellular carcinoma cells [87]. Although the exact mechanisms are not clear, miRNAs are identified as key factors in glucose and lipid metabolism [88]. Disruption in the ASB4-miRNA interaction could possibly play a role in the development of obesity and its related metabolic changes. ASB4 was also identified as a downstream protein of the transcription factor nuclear factor kappa b (NF-*κ*B) in the TNF-α signaling pathway [89]. Obesity is associated with the long-term low-grade inflammation of adipocytes and macrophages [90]. NF-*κ*B can initiate the inflammation cascade through the activation of macrophages [91], and this could be a new angle to examine the mechanistic relationships between ASB4 and obesity.

### 4.2. ASB4 and Vascular Development

During mouse early embryogenesis, the *Asb4* gene is highly expressed in the vasculature and spatiotemporally expressed in vasculogenic tissues such as yolk sac, dorsal aorta, and placenta [6]. In mice, embryonic day 7.5 (E7.5) embryos have low global *Asb4* mRNA expression, but at E9.5, *Asb4* expression is drastically increased, but begins to diminish after E10.5 and it is only expressed in tissues such as forelimb and umbilical vessels [6]. The onset of placental blood flow increases oxygen tension and triggers factors inhibiting HIF1α (FIH) to hydroxylate ASB4 at Asn 246, thereby promoting vascular development [6]. As the blood vessels mature, ASB4 mRNA expression in the highly vascularized organs (liver, lungs, and kidneys) is quickly downregulated and becomes undetectable [6]. In addition, *Asb4* gene transcription can be decreased by laminar shear stress and hypoxia insult [89], further highlighting the importance of ASB4 in the vascular system. In adults, *Asb4* mRNA expression was only detected in the testes, ovaries, and heart [6], suggesting that ASB4 is more important for early vascular development rather than for tissue maintenance. 

### 4.3. ASB4 and Preeclampsia

#### 4.3.1. ASB4, Implantation, and Preeclampsia

ASB4 is known to contribute to the differentiation of trophoblast stem cells into vascular cells and giant trophoblasts, which are necessary for embryo implantation and embryogenesis in mice [7]; however, its role in human pregnancy is unknown. Insufficient placentation resulting from a lack of ASB4 displays immature vascular patterning and retains the expression of placental progenitor markers, including ID2 [7]. ID2 is a member of the anti-differentiation ID protein family, which shares significant structural similarity to the basic helix-loop-helix (bHLH) family of transcription factors but lacks the basic domain [92]. Through heterodimerization with functional factors, ID2 blocks the transcription of pro-differentiation elements by preventing bHLH dimerization and subsequent translocation into the nucleus [93]. ASB4 mediates the ubiquitination and proteasomal degradation of ID2, and these processes are essential for the differentiation of cells [7]. Thus, ASB4 is a key element in trophoblast differentiation, implantation, and placentation. *Asb4^−^*^/*−*^ female mice develop mild preeclampsia-like phenotypes during pregnancy, including increased systolic blood pressure (SBP) and urinary albumin excretion, as well as a decreased litter size [7]. Because a loss of ASB4 leads to impaired trophoblast cell differentiation [7], it is highly possible that *Asb4^−^*^/*−*^ dams have defective implantation. However, neither blastocyst function nor endometrial receptivity—which could contribute to defective implantation—have been investigated in *Asb4^−^*^/*−*^ dams.

In addition, we found that circulation VEGF levels are decreased in *Asb4^−^*^/*−*^ dams, suggesting that imbalanced angiogenesis contributes to the pathogenesis of preeclampsia in these mice [8]. The mechanism by which lacking ASB4 leads to decreased VEGF is not clear. We hypothesize that increased ID2 may play a role because hepatocellular carcinoma (HCC) over-expressing ID2 showed decreased VEGF secretion [94]. However, neuroblastoma cells transfected with Ad-Id2 had increased VEGF expression [95]. Therefore, future study is needed to elucidate the effects of ID2 on VEGF in placentas lacking ASB4.

ID2 is expressed by human vascular endothelial and smooth muscle cells, and thus the dysregulation of this protein could cause maternal endothelial dysfunction. It would be interesting to know the levels of ID2 in vasculatures of *Asb4^−^*^/*−*^ mice and if they exhibit any dysfunctions. 

#### 4.3.2. ASB4, Obesity, and Preeclampsia

Recently, we demonstrated that HFD-induced obesity worsens the preeclampsia-like phenotypes in *Asb4^−^*^/*−*^ dams while the same diet treatment did not have obvious adverse effects on WT mice [9]. Therefore, we focused on comparing *Asb4^−^*^/*−*^ mice on a normal chow (NC) or HFD. *Asb4^−^*^/*−*^ female mice were started on a HFD (42% calories from fat) at an age of 21 days for five weeks. At an age of eight weeks, *Asb4^−^*^/*−*^ males were introduced for mating, and the female mice were fed a HFD throughout the entire pregnancy. We demonstrated the following: The HFD induced obesity in *Asb4^−^*^/*−*^ dams as evidenced by approximately 2× increase in visceral white fat mass in *Asb4^−^*^/*−*^ mice fed HFD compared to *Asb4^−^*^/*−*^ mice fed a normal diet. *Asb4^−^*^/*−*^ dams also had increased plasma cholesterol (1.6 ×), insulin (2.9 ×), and leptin (1.5 ×) levels compared to NC-fed *Asb4^−^*^/*−*^ dams while the plasma levels of glucose and triglyceride were not different between the two groups of mice. Plasma VEGF levels were decreasing in HFD-fed *Asb4^−^*^/*−*^ dams along with pregnancy, although these mice had higher plasma VEGF levels than NC-fed *Asb4^−^*^/*−*^ mice before pregnancy. Importantly, HFD-fed *Asb4^−^*^/*−*^ dams had higher blood pressure and urinary albumin excretion than NC-fed *Asb4^−^*^/*−*^ dams while HFD-fed *Asb4^−^*^/*−*^ dams had a decreased number of surviving fetuses compared to NC-fed *Asb4^−^*^/*−*^ dams at the term (e.g., 18.5 days post coitus), suggesting that all the preeclampsia-like phenotypes presented in *Asb4^−^*^/*−*^ dams were aggravated by HFD-induced obesity (Table 1). 

Following this observation, we investigated the role of altered factors in the increased severity of preeclampsia. We demonstrated that placental ID2 was higher in HFD-*Asb4^−^*^/*−*^ dams than NC-*Asb4^−^*^/*−*^ dams and investigated the role of insulin and leptin in increased placental ID2. In cultured HTR8/SVneo cell, high insulin (10 nM, comparable to the concentration observed in patients with insulin resistance [51,52]) increased ID2 protein levels while leptin (5–500 ng/mL) did not in these cells. These data suggest that hyperinsulinemia could play a role in increased ID2, leading to placental problems and decreased VEGF levels. Of note, we did not measure free fatty acid levels in plasma, but we cannot exclude the effects of free fatty acid on ID2 and placentation. The decreased placental VEGF could be due to increased ID2; however, this hypothesis needs to be tested.

## 5. Conclusions

Preeclampsia remains a significant and complex challenge in maternal and fetal medicine. Based on our discussion of preeclampsia, its association with obesity, and the roles of insulin, leptin, and ASB4 in these contexts, it is evident that these factors interact in complex and significant ways that influence both the maternal (the receiving endometrium) and fetal (the implanting blastocyst) side of pregnancy. 

## 6. Future Directions

Both environmental and genetic factors contribute to the development of obesity. Environmental factors, such as a high-calorie diet, promote obesity, while genetic factors can increase susceptibility to this condition. The manifestation of obesity is influenced by multiple genetic factors and their complex interactions with each other. A better understanding of the natural relationship between obesity and preeclampsia is needed, as these conditions not only adversely affect the mother and fetus during pregnancy but also have long-term health impacts.

Feeding *Asb4^−^*^/*−*^ mice with HFD (that is adjusted to diets consumed in Western societies) effectively replicates this human condition (obese combining with preeclampsia). This excellent mouse model provides a platform to better understand the molecular and genetic pathways involved in the pathogenesis of preeclampsia, including impaired uterine receptivity, trophoblast biology, and the implantation process, which subsequently leads to insufficient placentation and maternal endothelial dysfunction. This mouse model could also be used to test the efficacy of drugs targeting the signaling pathways altered by obesity, including insulin, leptin, and free fatty acid. Future research could aim to elucidate the precise mechanisms linking obesity-related metabolic dysregulations with preeclampsia pathogenesis, as well as to explore novel targets such as ASB4 that could potentially mitigate the impact of maternal obesity on pregnancy outcomes. By addressing these gaps, there is potential to improve maternal and fetal health and reduce the burden preeclampsia brings to the global healthcare systems.

## Figures and Tables

**Figure 1 ijms-25-09017-f001:**
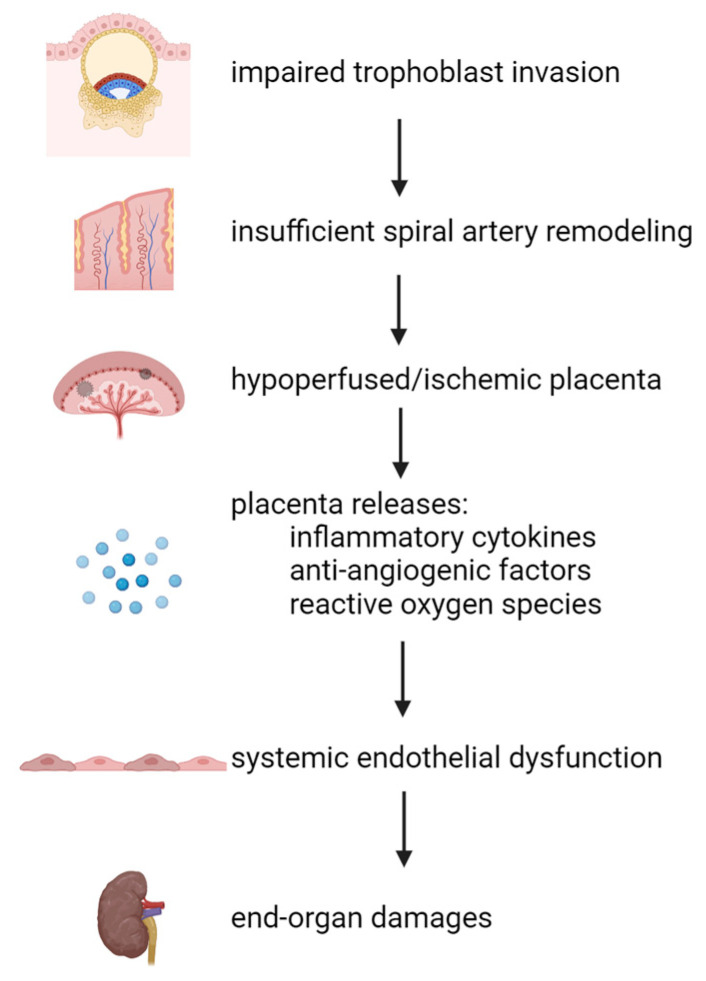
Defects during trophoblast invasion cause the shallow remodeling of uteroplacental spiral arteries. Abnormal vascular remodeling contributes to poor placentation due to the reduction in placental perfusion, making the placenta hypoxic/ischemic. The abnormal placenta releases inflammatory cytokines, anti-angiogenic factors, and reactive oxygen species into systemic circulation. These factors subsequently cause systemic endothelial dysfunction, leading to the clinical manifestations of preeclampsia (end-organ damage, proteinuria, and hypertension). “↓” denotes causation. All images are created with BioRender.com.

**Figure 2 ijms-25-09017-f002:**
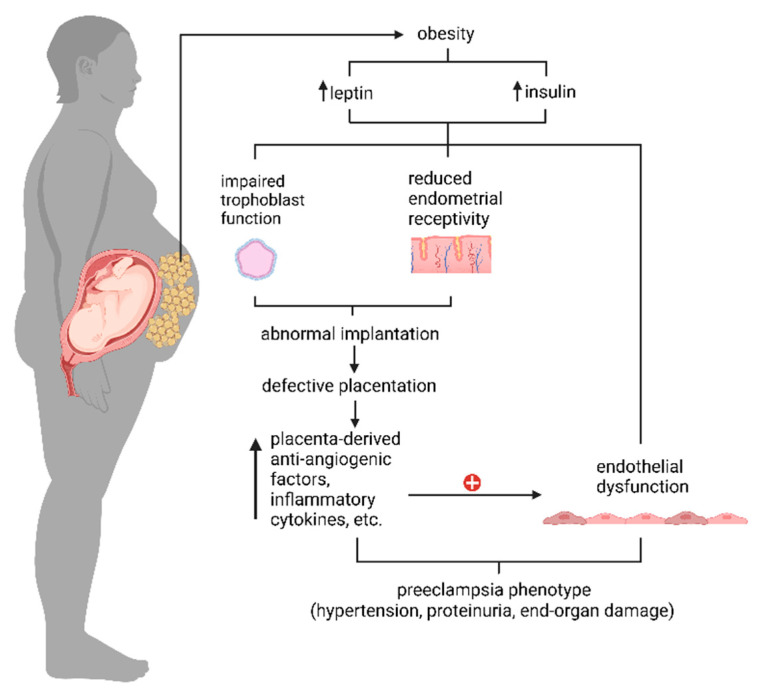
A graphical overview of how maternal obesity caused by excessive adipose tissue (denoted as yellowish spheres around the abdominal area of the pregnant mother) contributes to the pathogenesis of preeclampsia. All images are created with BioRender.com.

**Table 1 ijms-25-09017-t001:** HFD-induced obesity worsens preeclampsia-like phenotypes in *Asb4^−^*^/^*^−^* dams.

	Visceral Adipose Tissue (g)	P-Cholesterol (mg/dL)	P-Triglyceride (mg/dL)	P-Glucose (mg/dL)	P-Insulin (mg/dL)	P-Leptin (mg/dL)	Fetal Number (#)	Fetal Weight (g)	Placental Weight (g)
**NC-WT**	-	-	-	-	-	-	-	-	-
**HFD-WT**	-	-	-	-	-	↑ [74]	-	-	-
**NC-** ** *Asb4^−^* ** ** ^/^ ** ** * ^−^ * **	-	-	-	-	-	-	↓	-	↑
**HFD-** ** *Asb4^−^* ** ** ^/^ ** ** * ^−^ * **	↑	↑	-	-	↑	↑↑	↓↓	-	↑

NC: Normal Chow; HFD: High-Fat Diet. “↑” denotes increased level compared to NC-WT. “↓” denotes decreased level compared to NC-WT. “-” denotes no change in level compared to NC-WT. Double arrows denote increase or decrease to a greater extent [9].

## Data Availability

Not applicable.

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
