# Peer review of "The Roles of Obesity and ASB4 in Preeclampsia Pathogenesis"

_ijms, 2024, doi:10.3390/ijms25169017_

Round 1

Reviewer 1 Report

Comments and Suggestions for Authors

The review paper entitled “The Roles of Obesity and ASB4 in Preeclampsia Pathogenesis” by 

Yuye Wang et al. presents an interesting and clear overview about the role of the ASB4 protein in obesity and obesity-related preeclampsia. Provided the unpredictable nature of this serious life-threatening condition, this kind of basic research is much needed and very interesting.

Authors essentially provide the scientific background leading them to develop a mice model to study the role of the ASB4 and how they successfully worked to provide evidence that indeed ASB4 can have a crucial role in obesity and then in preeclampsia development.

Their presentation, however, is backward: starting from preeclampsia and going to ASB4.

In order to improve the focus and the reader's insight we would suggest starting the other way around.

An additional Figure depicting their mice model, the multi-faceted role of ASB4 and how the model can be useful to study obesity and then preeclampsia could be helpful.

Author Response

Comment: 

The review paper entitled “The Roles of Obesity and ASB4 in Preeclampsia Pathogenesis” by 

Yuye Wang et al. presents an interesting and clear overview about the role of the ASB4 protein in obesity and obesity-related preeclampsia. Provided the unpredictable nature of this serious life-threatening condition, this kind of basic research is much needed and very interesting.

Authors essentially provide the scientific background leading them to develop a mice model to study the role of the ASB4 and how they successfully worked to provide evidence that indeed ASB4 can have a crucial role in obesity and then in preeclampsia development.

Their presentation, however, is backward: starting from preeclampsia and going to ASB4.

In order to improve the focus and the reader's insight we would suggest starting the other way around.

An additional Figure depicting their mice model, the multi-faceted role of ASB4 and how the model can be useful to study obesity and then preeclampsia could be helpful.

Reply:

Thank you for spending time and effort reviewing our article. As you’ve suggested in the comment, we agree that it’s important to have an additional figure to provide more information about the ASB4-null mouse model. Therefore, we made a new table (Table 1.) and included it in the revised manuscript. Please see it on page 8. For the presentation order, we perfectly understand that starting from ASB4 can better grab the reader’s attention. However, ASB4 is only one of the many molecules that are involved in preeclampsia, and we believe that to better understand the roles of ASB4, it is necessary to understand the big picture first (what is preeclampsia, its relationship with obesity, etc..).

Reviewer 2 Report

Comments and Suggestions for Authors

The manuscript addresses an important problem. It is well-written and based on the latest scientific data. I have no significant comments, only minor ones:

Page 5 – Quercetin is a flavonoid occurring in food and is also available as a supplement. Simvastatin is a statin commonly used in clinical practice to lower cholesterol. Please change the following sentences in the manuscript.

Page 6 - Please check the number of amino acids in human leptin.  

Author Response

Comment:

The manuscript addresses an important problem. It is well-written and based on the latest scientific data. I have no significant comments, only minor ones:

Page 5 – Quercetin is a flavonoid occurring in food and is also available as a supplement. Simvastatin is a statin commonly used in clinical practice to lower cholesterol. Please change the following sentences in the manuscript.

Page 6 - Please check the number of amino acids in human leptin.  

Reply:

Thank you for spending time and effort reviewing our article. We agree that it is important to provide more information about quercetin and simvastatin, so the readers understand what they are used for. We have revised this sentence as you suggested and included it in the revised manuscript. For the number of amino acids in human leptin, it should be 167, not 146. We apologize for the mistake and thank you for pointing it out. Please see the yellow highlighted parts on page 4 (last word) and page 5.

Reviewer 3 Report

Comments and Suggestions for Authors

I have received for review a manuscript entitled “The Roles of Obesity and ASB4 in Preeclampsia Pathogenesis” which is being processed for publication in International Journal of Molecular Sciences.

I would like to congratulate the collective of authors for the proposed manuscript. The proposed case is an extremely interesting one, with potential therapeutic value.  Authors should pay attention to the following aspects in order to improve the proposed manuscript:

1. As a general reminder, the manuscript does not follow the journal's typesetting rules - missing key words, missing article identifiers on the left side of the first page, missing the font of the figures

2. The authors should mention the purpose of the article not only in the abstract, but also in the content of the manuscript.

3. Figures are very suggestive and make the manuscript easier to read, but they should be positioned in such a way that there are no empty pages

4.  Sections should be numbered

5. ASB4 - I suggest to the authors to make a table in a systematized way with the existing studies so far and the results obtained

6. Future research directions section should be separated.

In conclusion, the proposed manuscript brings to attention an extremely interesting topic, presenting scientific information with therapeutic value. The quality of the manuscript will be improved if the authors take into account the remarks made above.

Author Response

Reply:

Thank you for spending time and effort reviewing our article. We really appreciate the feedback you provided.

  1. As a general reminder, the manuscript does not follow the journal's typesetting rules - missing key words, missing article identifiers on the left side of the first page, missing the font of the figures

Thank you for pointing this out. We have added the keywords, identifier and revised the format.

  1. The authors should mention the purpose of the article not only in the abstract, but also in the content of the manuscript.

We have added an introduction paragraph that contains the purpose and background information of the topics we discussed.

  1. Figures are very suggestive and make the manuscript easier to read, but they should be positioned in such a way that there are no empty pages

This could be caused by editing which altered our original format, and we will work with the editor closely to make sure the figures are formatted and positioned correctly in the updated version of our manuscript.

  1. Sections should be numbered

We have added the numbers. Thank you.

  1. ASB4 - I suggest to the authors to make a table in a systematized way with the existing studies so far and the results obtained

We have added a table (Table 1. page 8)

  1. Future research directions section should be separated.

We have separated the original “conclusion and future direction” section.